# Elevation of Pro-Inflammatory Cytokine Levels Following Intra-Articular Fractures—A Systematic Review

**DOI:** 10.3390/cells10040902

**Published:** 2021-04-14

**Authors:** That Minh Pham, Julie Ladeby Erichsen, Justyna Magdalena Kowal, Søren Overgaard, Hagen Schmal

**Affiliations:** 1Department of Orthopedics and Traumatology, Odense University Hospital, 5000 Odense, Denmark; erichsenjulie@hotmail.com (J.L.E.); soeren.overgaard@rsyd.dk (S.O.); hagen.schmal@uniklinik-freiburg.de (H.S.); 2Department of Clinical Research, University of Southern Denmark, 5000 Odense, Denmark; 3Department of Orthopedic Surgery and Traumatology, Kolding Hospital, University Hospital of Lillebaelt, 6000 Kolding, Denmark; 4Molecular Endocrinology Laboratory (KMEB), Department of Endocrinology and Metabolism, Odense University Hospital, University of Southern Denmark, 5000 Odense, Denmark; jm.kowal@outlook.com; 5Department of Orthopaedic Surgery and Traumatology, Copenhagen University Hospital, 2400 Bispebjerg, Denmark; 6Department of Clinical Medicine, Faculty of Health and Medical Sciences, University of Copenhagen, 1165 Copenhagen, Denmark; 7Clinic of Orthopedic Surgery, Medical Center—University of Freiburg, Faculty of Medicine, University of Freiburg, 79085 Freiburg, Germany

**Keywords:** inflammatory, cytokines, biomarkers, intra-articular fracture, cartilage, joint injury, synovial fluid, osteoarthritis

## Abstract

**Introduction:** Intra-articular fractures are a major cause of post-traumatic osteoarthritis (PTOA). Despite adequate surgical treatment, the long-term risk for PTOA is high. Previous studies reported that joint injuries initiate an inflammatory cascade characterized by an elevation of synovial pro-inflammatory cytokines, which can lead to cartilage degradation and PTOA development. This review summarizes the literature on the post-injury regulation of pro-inflammatory cytokines and the markers of cartilage destruction in patients suffering from intra-articular fractures. **Methods:** We searched Medline, Embase, and Cochrane databases (1960–February 2020) and included studies that were performed on human participants, and we included control groups. Two investigators assessed the quality of the included studies using Covidence and the Newcastle–Ottawa Scale. **Results:** Based on the surveyed literature, several synovial pro-inflammatory cytokines, including interleukins (IL)-1β, IL-2, IL-6, IL-8, IL-12p70, interferon-y, and tumor necrosis factor-α, were significantly elevated in patients suffering from intra-articular fractures compared to the control groups. A simultaneous elevation of anti-inflammatory cytokines such as IL-10 and IL-1RA was also observed. In contrast, IL-13, CTX-II, and aggrecan concentrations did not differ significantly between the compared cohorts. **Conclusions:** Overall, intra-articular fractures are associated with an increase in inflammation-related synovial cytokines. However, more standardized studies which focus on the ratio of pro- and anti-inflammatory cytokines at different time points are needed.

## 1. Introduction

Post-traumatic osteoarthritis (PTOA) is a leading cause of chronic pain and disability worldwide [1,2]. The most common injury leading to PTOA is an intra-articular fracture, especially in the lower extremities [3,4,5]. The current gold-standard treatment of an intra-articular fracture is surgical stabilization and restoration of anatomy [6,7,8]. However, despite correct restoration of joint congruency, the risk of PTOA following ankle fractures 10 years after injury remains higher than 40% [9,10].

Previous studies indicated that joint injuries initiate an inflammatory cascade, changing the synovial cytokine composition [11,12]. The overexpression of pro-inflammatory cytokines can lead to cartilage degradation and subsequent PTOA development [2,13,14], a fact which, to date, has been largely ignored in standard fracture treatment.

Joint injuries involving damage to the anterior cruciate ligament (ACL) or meniscus were accompanied by elevated levels of pro-inflammatory cytokines (interleukin (IL)-1β, IL-6, IL-8, and tumor necrosis factor-α (TNF-α)), matrix-degrading enzymes (matrix metalloproteinase (MMP)-1, MMP-3, and MMP-9), and suppression of anti-inflammatory cytokines (IL-1RA and IL-10) compared to uninjured control groups [15,16,17]. These cytokines and proteolytic enzymes originate from cartilage, synovium, meniscus, or bone and participate in the initiation of cartilage degradation, leading to PTOA [18].

Previous research has mainly focused on soft tissue knee injuries rather than on intra-articular fractures. It is expected that the high energy impact causing an intra-articular fracture is even more severe than in a meniscus or ACL injury. This is likely to lead to a greater inflammatory response and may be more damaging for the cartilage [19]. Very little is currently known about the composition of synovial cytokines following intra-articular joint fractures; however, it is crucial to identify these potential cartilage-damaging cytokines to optimize the current treatment and to provide options to modulate the inflammatory response.

In this study, we performed a systematic review of the literature, focusing on synovial pro-inflammatory cytokine regulation following intra-articular fractures. The objective of this study was to investigate which cytokines are elevated in patients with fractures involving the articular surface compared to control groups. More specifically, we were interested in the following proteins: aggrecan (ACG), C-terminal telopeptides of type Ⅱ collagen (CTX-II), basic fibroblast growth factor (bFGF), interferon-y (IFN-y), IL-1α, IL-1β, IL-1RA, IL-2 IL-4, IL-6, IL-8, IL-10, IL-12p70, IL-13, MMP-1, MMP-3, MMP-9, transforming growth factor-β1 (TGF-β1), and TNF-α.

## 2. Materials and Methods

### 2.1. Protocol and Registration

This systematic review was planned, conducted, and reported according to the PRISMA (Preferred Reporting Items for Systematic Reviews and Meta-Analysis) Statement [20]. The study protocol was registered at the PROSPERO International Prospective Register of Systematic Reviews (CRD42019126857) before data extraction and data analysis began.

### 2.2. Search Strategy

The following databases were included in the literature search: Cochrane, Medline, and Embase. A research librarian at our university assisted with the development of the search strategy. The review followed the guidelines of the PICO (population, intervention, comparison, and outcomes) model according to the Oxford Centre for Evidence-Based Medicine (CEBM). We included studies up to 1 February 2020 that were written in English, German, or a Scandinavian language and included human participants. We used the following search terms: (intra-articular fracture) AND (biomarkers OR cytokines OR synovial fluid). The search strategy is attached as Supplemental Material 1. The outcome measure was the difference in protein concentrations (ACG, CTX-II, bFGF, IFN-y, IL-1α, IL-1β, IL-1RA, IL-2 IL-4, IL-6, IL-8, IL-10, IL-12p70, IL-13, MMP-1, MMP-3, MMP-9, TGF-β1, and TNF-α) between acute fractured joints and controls. Both absolute concentrations and ratios were analyzed.

### 2.3. Study Selection and Data Collection Process

Management of the search results was carried out in Covidence (www.covidence.org, (accessed on 27 June 2017) Veritas Health Innovation). Duplicate studies were identified and removed. The titles and abstracts of all retrieved studies (*n* = 2238) were reviewed. The two reviewers (TMP/JLE) independently analyzed the title and abstract of each article and determined its eligibility. All of the eligible studies (*n* = 15) were read in full, and, of these, a further nine studies were excluded (Figure 1). Data extraction from the remaining six studies was performed using a data extraction sheet based on the type of study, the type of participants, the type of intervention, and the outcomes. The corresponding authors were contacted if data were missing. The data were extracted and double-checked (TMP/JLE). Disagreements between the two reviewers were resolved by discussion. If no agreement could be reached, a third reviewer (HS) was consulted.

The following information was extracted from each study: (1) the characteristics of the study (author, year, study design, method, and fracture joint), (2) control group, (3) outcome measures (cytokine levels or ratios, concentration units), and (4) significance levels.

### 2.4. Risk of Bias in Individual Studies

The Newcastle–Ottawa Scale, developed for quality assessment of cross-sectional studies, was used to determine the bias for all of the studies included in this review (Supplemental Material 2).

### 2.5. Quality Assessment

All included studies were cross-sectional. Therefore, the Newcastle–Ottawa Scale was used to assess the quality of all six studies. Two studies [11,12] were scored good and very good. One study [21] scored satisfactory, while three studies [2,22,23] scored unsatisfactory (Table 1). The main cause of low scores was problems with comparability and selection. The three unsatisfactory studies [2,22,23] did not use the contralateral joint and instead used cadavers, OCD grade 2, or OA patients from another group as a control. Furthermore, two unsatisfactory studies [2,22] included less than 10 patients. The agreement between the two observers (TMP/JLE) for this evaluation was good, and any disagreements were resolved after discussion.

## 3. Results

Six studies fulfilled the inclusion criteria. Four studies [2,11,21,22] retained synovial fluid for analysis from patients with ankle fractures, one from patients with knee fractures, and one from patients with elbow fractures. The studies were carried out between January 2014 and December 2019. As a control, three studies [11,12,21] used the contralateral joint, and the remaining three studies [2,22,23] used cadavers, osteoarthritis (OA), and osteochondritis dissecans (OCD) patients. The time points for synovial fluid aspiration after injury varied from 0 to 40 days (Table 1).

Five out of six studies [2,11,12,21,23] used saline injection before aspiration of synovial fluid. While four studies [2,11,21,23] used urea or total protein levels to correct for dilution, two studies [12,22] did not provide detailed information on how absolute concentrations were calculated. All six studies used singleplex or multiplex enzyme-linked immunosorbent assays (ELISA) from different companies to determine cytokine concentrations. The sets of cytokines analyzed in these six studies differed considerably (Table 2). The absolute concentrations were only available in three studies [11,12,21]. After our request, only Schmal et al. [2] responded and provided additional data. Due to the lack of absolute cytokine concentrations from the remaining studies, we were unable to conduct a meta-analysis.

Overall, IL-1RA, IL-6, IL-8, IL-10, MMP-1, MMP-3, and MMP-9 were found to be elevated in all studies that included these cytokines. IL-1β and IFN-y were elevated in four out of five studies [11,12,21,22,23], whereas for IL-2, IL-12p70, and TNF-α, three out of four studies reported a significant elevation. CTX-II, ACG, and IL-13 were not shown to be significantly elevated in joints with fractures compared to the control groups (Table 1).

## 4. Discussion

In this review, we aimed to evaluate if intra-articular fractures are associated with an increase in inflammation-related synovial cytokines. We found a significant elevation of several pro-inflammatory cytokines, including IFN-y, IL-1β, IL-2, IL-6, IL-8, IL-12p70, TNF-α, and MMPs, in patients suffering from intra-articular fractures compared to control groups. We also found a simultaneous elevation of the anti-inflammatory cytokines IL-10 and IL-1RA.

Numerous studies identified IL-1β and TNF-α as principal mediators for an acute inflammatory response after joint trauma. Both cytokines are considered key players in the pathogenesis of OA [24,25]. IL-1β and TNF-α are upregulated by several cell types, including chondrocytes, cells forming the synovial membrane, and infiltrating inflammatory cells such as mononuclear cells. Upregulation of IL-1β and TNF-α stimulates cartilage matrix degradation by interfering with the synthesis of type II collagen and ACG and inducing a group of MMPs that have a destructive effect on cartilage components. Furthermore, upregulation of IL-1β and TNF-α in the synovial fluid also stimulates the synthesis of other pro-inflammatory cytokines, including IL-6 and IL-8.

Several studies in ACL patients have reported elevation of IL-1β and TNF-α [15,16,26], which is mainly in agreement with the findings of this systematic review in intra-articular fracture patients. Bigoni et al. [26] reported a concentration of IL-1β in ACL patients that was higher on average than in proximal tibia fracture patients [12,23]. In contrast, the concentration of TNF-α was higher following fractures than when following ACL injuries [12,23,26]. Furthermore, along with the studies in the present systematic review which found a significant increase in these cytokines, Schmal et al. [2] and Haller et al. [12] also reported a non-significant elevation of IL-1β and TNF-α, respectively, in fracture patients. While an ACL tear is a standardized injury, fractures differ in comminution and complexity. This might partially explain the discrepancies in the findings.

Bigoni et al. [26] found that IL-1β returned to normal levels after 3 days and remained low at the following measurement. The same finding was reported by Adams et al. [27] for ankle fractures 6 months after surgery.

In the present review, the concentration of synovial IL-6 and IL-8 in intra-articular fractures was reported to be increased in the studies, as found in previous studies investigating ACL lesions. However, the absolute concentrations of IL-6 and IL-8 in intra-articular fracture joints seem to be higher compared to ACL patients [16,26]. Production of IL-6 and IL-8 in the joint after an injury is usually in response to IL-1β and TNF-α. This production is mainly implemented by chondrocytes and macrophages and plays a major role in OA [25]. In contrast to the continued elevation of IL-1β and TNF-α, Adams et al. [27] reported that IL-6 and IL-8 remained elevated in ankle fracture patients after 6 months. Although this was the only study analyzing long-term time points, these high levels of IL-6 and IL-8 may indicate continuous inflammation and destruction of cartilage in the joint, leading to PTOA development.

The concentrations of anti-inflammatory cytokines such as IL-10 and IL-1RA increased simultaneously with other pro-inflammatory cytokines. IL-10 is a potent anti-inflammatory cytokine that has shown a chondroprotective effect in the course of OA by stimulating the synthesis of type II collagen and ACG [28]. Furthermore, IL-10 is involved in the inhibition of metalloproteinases and in chondrocyte apoptosis, as well as the downregulation of IL-1β and TNF-α, by stimulating the production of IL-1RA [25]. IL-1RA can bind to the IL-1R receptor, thereby blocking the connection to IL-1β and indirectly inhibiting the pro-inflammatory effect of IL-1β. In this systematic review, data for IL-1RA were only reported in patients with proximal tibia fractures [12]. However, the concentrations of IL-1RA in these fracture patients seem to be lower than previously reported in ACL studies [16]. This may indicate that the higher energy impact resulting in a fracture may cause a higher degree of catabolism and joint destruction.

MMPs are a family of endopeptidases that are critically important in extracellular matrix remodeling. MMP-1 cleaves collagen types 1, 2, 3, 7, and 10 and is thought to play an important role in the pathogenesis of OA by mediating cartilage and bone destruction. MMP-3, also called stromelysin-1, plays an active role in the degradation and reconstitution of the extracellular matrix and stimulates other latent-type MMPs such as MMP-9. Adams et al. [27] showed that concentrations of MMP-1, MMP-3, and MMP-9 were increased up to 100-fold in fractured joints compared to healthy joints. These findings were consistent with a previous study looking at patients with ACL injury [17]. However, no significant elevation of ACG and CTX-II concentrations was observed in the studies included in this systematic review. This inconsistent finding may be due to the timing of synovial fluid collection after injury or may reflect the limited number of included studies that analyzed these proteins.

These findings must be interpreted with caution because only six studies were included in this systematic review. Furthermore, two studies had less than ten patients [2,22], and only three studies used the contralateral joint for comparison [11,12,21]. The other three studies used cadavers, paired OCD patients, or unpaired OA patients in another joint as controls [2,22,23]. Another challenge for all of the included studies was the procedure of synovial fluid collection from the involved joints. In a normal joint, and even after articular fractures, only a limited amount of synovial fluid is present in the joint space. Hence, some authors rinsed the joint space with saline before aspiration and corrected the dilution factor using urea concentration or total protein levels. Not all studies contain a sufficient description of this procedure, which makes it difficult to compare and evaluate the results. Another issue is that all six studies used different periods for synovial fluid collection, ranging from 0 to 40 days. For example, the period for inclusion was 0–1 days in the study conducted by Haller et al. [12] but 8–40 days in the study by Adams et al. [11]. The concentrations of cytokines change continuously with time after injury, which makes it difficult to compare the measured concentrations of samples taken at different times. Furthermore, it is important to use consistent ELISA methods to allow a comparison of the level of cytokines across studies. In short, the concentrations of cytokines can be influenced by fracture complexity, the time between fracture and measurement, and methods of puncture and cytokine measurement used. A further limitation of this systematic review is that we decided not to include gray literature (unpublished data such as conference proceedings, papers, or posters) and included only studies in English, German, and the Scandinavian languages. However, we believe that the most relevant studies were included despite these exclusions.

The adapted Newcastle–Ottawa Scale for cross-sectional studies was used as a quality assessment for the six included studies. Although the characteristics of the studies differed substantially, three studies had acceptable quality, and the results of all trials suggest principally the same outcome: intra-articular fractures drive synovial inflammation.

To the best of our knowledge, no human clinical studies have examined the effect of supplementing anti-inflammatory therapy when treating acute fractures, as reported in a systematic review by Schmal et al. [29]. A study by Wang et al. [30], using intra-articular hyaluronic acid combined with conventional surgery, reported positive effects in ankle fracture patients. Intra-articular treatment with IL-1RA, an IL-1β antagonist, was reported by Kraus et al. [31] to have positive effects in patients with an ACL tear. However, IL-1RA showed no significant effects in patients with degenerative knee OA [32,33]. Unlike degenerative OA, in PTOA, we know the starting time point. Clinical intra-articular treatment with anti-inflammatory agents may be effective if applied in a timely manner after an intra-articular fracture. However, it is beyond the topic of this review.

### Suggestions for Future Work

To support the current findings, future studies should employ power calculations to allow statistically robust findings and use consistent, well-documented methods of synovial collection. Of the studies in this review, only Schmal et al. [2] provided a sampling method validation. It may be worthwhile to evaluate the most reliable method to correct for dilution during synovial collection. Another challenge to be addressed is the fluctuation of cytokine concentration over time. It could be interesting to identify the ratio of pro- and anti-inflammatory cytokines at given time points after injury.

## 5. Conclusions

Based on the surveyed literature, the synovial concentrations of IL-1β, IL-1RA, IL-2, IL-6, IL-8, IL-10, IL-12p70, IFN-y, MMP-1, MMP-3, MMP-9, and TNF-α were found to be significantly elevated following intra-articular fractures compared to different control groups. No difference was found for ACG, CTX-Ⅱ, and IL-13. Despite methodological differences in the included studies, these findings are supported by the results of all studies. The potential cartilage-degrading effect of the analyzed cytokines may be further examined to better determine the risk of later OA development.

## Figures and Tables

**Figure 1 cells-10-00902-f001:**
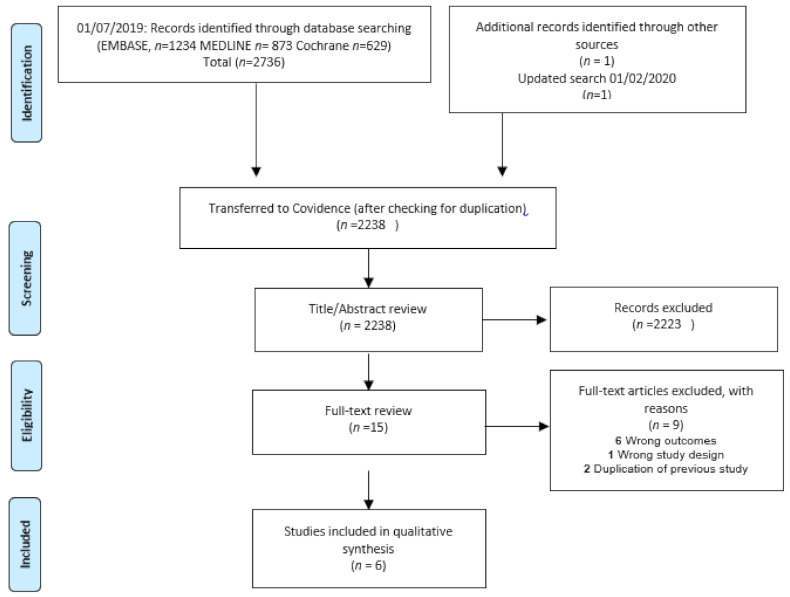
Study flow diagram.

**Table 1 cells-10-00902-t001:** Demographics and Results of the Included Studies (*n* = 6).

Year/Author/Country	Study Design	Fracture Joint	Control Group	Time of Synovial Fluid Collection (Days after Injury)	Non-Significant Outcomes (Fracture vs. Control)	Significant Outcomes (Fracture vs. Control)	Newcastle-Ottawa Scale (NOS)
**2015/Adams/USA (11)**	Cross-sectional	Ankle (*n* = 21)	Contralateral ankle (*n* = 21)	8–40 days	IL-1α, IL-2, CTX-II	**IFN-γ, IL-1β, IL-6, IL-8, IL-10, IL-12p70, MMP-1, MMP-3, MMP-9, TNF-α**	Good study (8 stars)
**2015/Furman/USA (22)**	Cross-sectional	Ankle (*n* = 6)	Knee OA (*n* = 6)	5–21 days	None	**IFN-γ, IL-1β, IL-6, IL-8, IL-10, IL-12p70, TNF-α**	Unsatisfactory study (6 stars)
**2014/Schmal/DE (2)**	Cross-sectional	Ankle (*n* = 8)	OCD grade 2 (*n* = 8)	0–4 days	ACG, IL-1β	**bFGF**	Unsatisfactory study (4 stars)
**2015/Haller/USA (12)**	Cross-sectional	Knee (*n* = 45)	Contralateral knee (*n* = 45)	0–1 day	IL-1α, IL-4, IL-12p70, IL-13, TNF-α	**IFN-y, IL-1β, IL-2, IL-6, IL-8, IL-10, IL-1RA**	Very good study (9 stars)
**2017/Godoy-Santos/BRA**	Cross-sectional	Ankle (*n* = 16)	Cadavers (*n* = 5)	2–5 days	IFN-y, TGF-β1	**IL-2, IL-6, and IL-10**	Unsatisfactory study (4 stars)
**2019/Wahl/ USA (21)**	Cross-sectional/Case-control	Elbow (*n* = 11)	Contralateral elbow (*n* = 11)	0–17 days	CTX-II	**IFN-y, IL-1 β, IL-2, IL-4, IL-6, IL-8, IL-10, IL-12p70, IL-13, MMP-1, MMP-3, MMP-9, TNF-α**	Satisfactory study (7 stars)

USA: United States of America; DE: Germany; BRA: Brazil; OA: osteoarthritis; OCD: osteochondritis dissecans; ACG: aggrecan; CTX-II: C-terminal telopeptides of type II collagen; bFGF: basic fibroblast growth factor; IFN-y: interferon-gamma; IL: interleukin; MMP: matrix metalloproteinase; TGF-β: transforming growth factor-beta; TNF-α: tumor necrosis factor-alpha. Cytokines in **bold** indicate a significant difference compared to the control group.

**Table 2 cells-10-00902-t002:** Absolute Concentrations of Cytokines in Synovial Fluid from Fracture and Control Joints.

Study		Pro-Inflammatory Cytokines											Anti-Inflammatory Cytokines		
Year/Author/Country	Joint Involved	IL-1α (pg/mL)	IL-1β (pg/mL)	IL-2 (pg/mL)	IL-6 (ng/mL)	IL-8 (ng/mL)	IL-12p70 (pg/mL)	TNF-α (pg/mL)	IFN-Y (pg/mL)	MMP-1 (ng/mL)	MMP-3 (ng/mL)	MMP-9 (ng/mL)	IL-4 (pg/mL)	IL-10 (pg/mL)	IL-1RA (pg/mL)
**2015/Adams/USA** [11]	Fractured ankle	1.81 ± 2.97 (ns)	**2.12 ± 2.90**	1.11 ± 2.26 (ns)	**1.83 ± 1.78**	**1.13 ± 1.07**	**1.06 ± 3.65**	**3.78 ± 3.56**	**0.44 ± 0.34**	**830.3 ± 247.2**	**1776.7 ± 629.1**	**38.8 ± 50.6**		**10.12 ± 12.95**	
	Contralateral ankle	2.44 ± 4.24	**0.29 ± 0.51**	0.31 ± 0.67	**0.12 ± 0.52**	**0.009 ± 0.016**	**0.18 ± 0.00**	**0.55 ± 1.04**	**0.37 ± 0.00**	**13.5 ± 37.5**	**59.7 ± 124.0**	**9.4 ± 20.9**		**0.33 ± 0.52**	
**2015/Furman/USA** [22]	Fractured ankle vs. knee OA		**Significant increase**		**Significant increase**	**Significant increase**	**Significant increase**	**Significant increase**	**Significant increase**					**Significant increase**	
**2014/Schmal/DE** [2]	Fractured ankle		18.7 ± 24.8 (ns)												
	OCD grade 2 ankle		10.9 ± 3.70												
**2015/Haller/USA** [12]	Fractured knee	Below LLOD	**1.9** **(1.2–2.8)**	**3.5** **(2.1–5.3)**	**3.1** **(1.4–6.7)**	**0.22 (0.14–0.36)**	5.4 (3.8–7.4) (ns)	9.6 (7.5–12.4) (ns)	**3.3** **(2.2–4.9)**				Below LLOD	**88.6** **(63.5–123.5)**	**113.6** **(68.7–187.5)**
	Contralateral knee		**0.8** **(0.4–1.3)**	**1.6** **(0.8–2.6)**	**0.006 (0.002–0.014)**	**0.004 (0.002–0.006)**	7.8 (5.6–10.7)	9.5 (7.3–12.1)	**1.7** **(1–2.6)**					**2.4** **(1.5–3.8)**	**12.6** **(7.3–21.4)**
**2017/Godoy-Santos/BRA** [23]	Fractured ankle vs. cadavers			**Significant increase**	**Significant increase**				Non- significant increase					**Significant increase**	
**2019/Wahl/USA** [21]	Fractured elbow		**4.47 ± 3.81**	**1.36 ± 0.93**	**1.76 ± 0.37**	**1.38 ± 0.80**	**5.62 ± 2.91**	**2.58 ± 0.60**	**8.37 ± 5.32**	**240 ± 430**	**1300 ± 910**	**280 ± 250**	**45.6 ± 93.7**	**2.89 ± 1.84**	
	Contralateral elbow		**0.17 ± 0.15**	**0.42 ± 0.7**	**0.004 ± 0.007**	**0.012 ± 0.009**	**0.07 ± 0.08**	**0.80 ± 0.82**	**0.33 ± 0.39**	**2.0 ± 2.0**	**100.0 ± 70.0**	**30.0 ± 40.0**	**0.10 ± 0.23**	**0.15 ± 0.29**	

USA: United States of America; DE: Germany; BRA: Brazil; OA: osteoarthritis; OCD: osteochondritis dissecans; IFN-y: interferon-gamma; IL: interleukin; MMP: matrix metalloproteinase; TGF-β: transforming growth factor-beta; TNF-α: tumor necrosis factor-alpha. The absolute cytokine concentrations in the fracture joints and control joints are presented as in the original data: mean *±* standard deviation or mean (confidence interval). LLOD: Lower limit of detection. Data in **bold** indicate a significant difference in cytokine levels between fracture and control joints. (ns): the fractured joint is not significantly different from the control joint.

## Data Availability

Data sharing not applicable. No new data were created or analyzed in this study. Data sharing is not applicable to this article.

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
