# Peer review of "Elevation of Pro-Inflammatory Cytokine Levels Following Intra-Articular Fractures—A Systematic Review"

_cells, 2021, doi:10.3390/cells10040902_

Round 1

Reviewer 1 Report

I think the initial idea from which this review started is very interesting. Intra-articular fractures are recognized as a very important cause of PTOA. The reason for this phenomenon is linked to the inflammatory environment in which some cytokines can cause cartilage degradation and in the long term PTOA development. The main problem of this review is that of having considered only studies including a control group, leading to draw conclusions on only 6 studies, heterogeneous from each other and with very low patient numbers. The control groups also varied between using samples from cadavers to samples obtained from the contralateral joint.
The conclusions reached by the authors are unfortunately not supported by sufficient literature to be generalized.
I believe that the study should be completely redone by examining the large mass of studies that report the evolution of the intraarticular inflammatory picture even in the absence of a control group.

Author Response

Point-by-point reply

We would like to thank the reviewer for taking their time to give constructive feedback on this manuscript. We have carefully addressed and answered all questions raised by reviewer which we believe has significantly improved the quality of the manuscript. A detailed point-by point replies to the questions raised, and the corresponding changes made in the manuscript, is provided below.

Enclosed please find our revised manuscript (clean version) and a tracked changes document.

The corresponding author,

That Minh Pham, MD.

Author reply:

Thank you the reviewer for the comment and for sharing the same interest in the inflammatory response after intra-articular fracture.  We decided to perform a systematic review on this topic, because we could not identify any systematic review reporting this inflammatory response after intra-articular fracture in the literature.

We had initially also considered to include other joint injuries such as the knee joints with meniscus or ACL lesion. However, the time of SF collection in this patient group was primary during the surgical repair, which may vary from days to months after injury. This would equally be difficult to compare to the cytokine levels found in fracture patients, which primary was collected within 14 days from injury. Furthermore, the heterogeneity in the methodology would still make it difficult to compare the results. In addition, a requirement of a control group was considered important, because the cytokines level in normal joints are not persistent and well descripted. It might also be difficult to determine the degree of cytokines elevation without a comparison.  Consequently, we decided to perform a systematic review with a clear research question only focusing on intra-articular fracture. We do agree that only six studies were included in the final qualitative analysis and that might not be a lot, however, it is most likely what exist in the literature.

To the best of our knowledge, this is the first study using systematic and reproducible methods to identify, select and critically appraise the relevant research. Furthermore, this study identified the current challenges in the methods of SF collection, which would be useful in future studies.

Reviewer 2 Report

In the study presented here, the authors aimed to summarizes the literature on the post-injury regulation of pro-inflammatory cytokines and the markers of cartilage destruction in patients suffering from intra-articular fractures. They found that intra-articular fractures are associated with an increase in inflammation-related synovial cytokines. The observations are clear.

Major Questions:

  1. Page 2, Line 45: The references of this sentence “Previous studies indicated that…” should be provided.

  1. Page 8, Line 1-11: The “2.5. Quality assessment” should move to Page 3, Line 114.

Author Response

Point-by-point reply

We would like to thank the reviewer for taking their time to give constructive feedback on this manuscript. We have carefully addressed and answered all questions raised by reviewer which we believe has significantly improved the quality of the manuscript. A detailed point-by point replies to the questions raised, and the corresponding changes made in the manuscript, is provided below.

Enclosed please find our revised manuscript (clean version) and a tracked changes document.

The corresponding author,

That Minh Pham, MD.

Major Questions:

  1. Page 2, Line 45: The references of this sentence “Previous studies indicated that…” should be provided.

 Author reply:

Thank you for your comments. We do agree and additional references will be added in the manuscript.

Author action:

As request, two following references are added  (Page 2, Line 46)

  1. Adams SB, Setton LA, Bell RD, et al. Inflammatory cytokines and matrix metalloproteinases in the synovial fluid after intra-articular ankle fracture. Foot Ankle Int 2015; 36: 1264-71.

  1. Haller JM, McFadden M, Kubiak EN, et al. Inflammatory cytokine response following acute tibial plateau fracture. J Bone Joint Surg Am 2015; 97: 478-83.

  1. Page 8, Line 1-11: The “2.5. Quality assessment” should move to Page 3, Line 114.

Author reply:

Thank you for paying intention to this detail. We do agree that this section could easily be removed to Page 3, Line 114.  However, we had initially placed it under results section, because this quality assessment was considered as a result. We would be equally happy to remove this section. In this matter, we would let the editor decides what is best according the structure of this journal.

Reviewer 3 Report

The manuscript is a systematic review examining publications inflammatory markers in the synovial fluid after intraarticular fractures. The methodology of the manuscript is solid. The authors discuss all potential shortcomings of their work. They only found three high quality paper. Severity and site of the injuries varied. Time of sample collection after injuries differed, too. The papers used different controls: contralateral joint, osteoarthritis sample, osteochondritis dissecans sample and postmortem sample. The studies investigated different inflammatory cytokines. It would be nice to find papers that report synovial cytokine concentrations after non-intraarticular fractures. This would clarify if elevated cytokine levels are specific to intraarticular fractures. However, I could not find any such papers, so I would not persuade the authors to try. I would suggest one small correction. The authors state the no anti-inflammatory medication is used in conjunction with fractures. I think NSAD drugs are routinely administered and they are antiphlogistic. Potential effect of NSAIDS could be discussed. Alternatively, outcome of such fractures could be assessed in patients that accidentally undergo immunosuppressive therapy. But that would be the topic of another work.

Author Response

Point-by-point reply

We would like to thank the reviewer for taking their time to give constructive feedback on this manuscript. We have carefully addressed and answered all questions raised by reviewer which we believe has significantly improved the quality of the manuscript. A detailed point-by point replies to the questions raised, and the corresponding changes made in the manuscript, is provided below.

Enclosed please find our revised manuscript (clean version) and a tracked changes document.

The corresponding author,

That Minh Pham, MD.

Author reply:

Thank you the reviewer for the comment and for sharing the same interest in the inflammatory response after intra-articular fracture.  We decided to perform a systematic review on this topic, because we could not identify any systematic review reporting about the inflammatory response after intra-articular fracture in the literature.

We had initially also considered to include non-intra-articular fracture such as the knee joints with meniscus or ACL lesion. However, the time of SF collection in this patient group was primary during the surgical repair, which may vary from days to months after injury. This would equally be difficult to compare to the cytokine levels found in intra-articular fracture patients, which primary was collected within 14 days from injury. Consequently, we decided to perform a systematic review with a clear research question only focusing on intra-articular fracture.

We do agree that only six studies were included in the final qualitative analysis and that might not be a lot, however, it is most likely what exist in the literature. To the best of our knowledge, this is the first study using systematic and reproducible methods to identify, select and critically appraise the relevant research, and may justify to be published.  Furthermore, this study identified the current challenges in the methods of synovial fluids collection, which would be useful in future studies.

Concerning the use of nonsteroidal anti-inflammatory drugs (NSAIDs) in intra-articular fracture treatment as anti-inflammatory medication is very interesting. It was suggested in early literature, that NSAID, due to the anti-inflammatory ability, might delay fracture-healing post-surgery.  Despite, several systematic reviews had subsequently reported no association of NSAID and fracture healing (DOI: 10.1007/s00167-012-2095-2, DOI: 10.1007/s00167-012-2095-2), many surgeons still hesitate using NSAID after surgery. We do agree that NSAID do have antiphlogistic ability and is wildly used as anti-inflammatory medication.  It could be interesting to examine if routinely intake of NSAID would reduce the initial inflammatory cascade and subsequently reduce the risk of PTOA development. However, as specified in Page 10, Line 117-119, we are not aware about any human study that had evaluate the effect of NSAID after intra-articular fracture and the association of PTOA. Nevertheless, we are likely not updated. In case the reviewer know of such study, we would appreciate to hear about it, because it would probably improve the quality of our manuscript.   

Round 2

Reviewer 1 Report

I continue to have major problems regarding the significance of this review. The studies that are analyzed are only 6, very heterogeneous by type of fracture and although all with a control group, these control groups were also heterogeneous between the studies. The only constant parameter is the time elapsed since the injury. I understand that these are the only studies with the characteristics selected by the authors, but I believe that it would have been necessary to at least compare these results with those of the more than two thousand publications that have instead been excluded. Although certainly very heterogeneous from each other, these thousands of studies are far more informative and, on the whole, more credible than the 6 studies considered by the authors.

The reviews are published to provide a key to understanding the problem analyzed, but if, as in this case, they reduce the literature to an unrepresentative number of researches, they lose their value. I think this review should be rewritten to compare the results of the 6 studies with those of the wider literature on the same topic.

Reviewer 2 Report

Page 8, Line 1-11: The “2.5. Quality assessment...” should move to Page 3, Line 121.

Reviewer 3 Report

The manuscript is fit for publication.